

# Hydroxyl airglow observations for investigating atmospheric dynamics: results and challenges

Sabine Wüst[1], Michael Bittner[1,2], Patrick J. Espy[3], W. John R. French[4], Frank J. Mulligan[5]

[1] Erdbeobachtungszentrum, Deutsches Zentrum für Luft- und Raumfahrt Oberpfaffenhofen, 82234 Wessling, Germany
[2] Institut für Physik, Universität Augsburg, 86159 Augsburg, Germany
[3] Department of Physics, Norwegian University of Science and Technology, Trondheim, Norway
[4] Australian Antarctic Division, 203 Channel Hwy, Kingston, Tasmania, 7050, Australia
[5] Department of Experimental Physics, Maynooth University, Maynooth, Co. Kildare, Ireland

*Correspondence to*: Sabine Wüst (sabine.wuest@dlr.de)

**Abstract.** Measurements of hydroxyl (OH*) airglow intensity are a straightforward and cost-efficient method which allows information to be derived about the climate and dynamics of the upper mesosphere / lower thermosphere (UMLT) on different spatiotemporal scales during darkness.

Today, instrument components can be bought "off-the-shelf" and developments in detector technology allows operation without cooling or at least without liquid nitrogen cooling, which is difficult to automate. This makes instruments compact
and suitable for automated operation.

Here, we provide an overview of the scientific results regarding atmospheric dynamics and relying on long-term ground-based OH*-airglow measurements or airglow measurements using a network of ground-based instruments. It includes further results from global or nearly-global satellite-based OH*-airglow measurements. Additionally, the results from the very few available airborne case studies using OH*-airglow instruments are summarised. Scientific and technical challenges for the next few
years are described.

## 1 Introduction

At present, nocturnal hydroxyl (OH*) airglow measurements are performed from the ground, aircraft, balloon and space. Ground-based OH* airglow observations are far more cost effective than measurements from aircraft and satellite. They have been made since the 1950's and are well suited to long-term studies if calibrations are well maintained, and changes or
replacements in the measurement equipment over the years are handled with care and are well documented. More recently, satellite-based and airborne measurements have been carried out that are not severely affected by clouds in most cases. The former provide global or nearly global data, the latter the greatest spatial flexibility. Balloon-borne measurements, as presented recently by Wang et al. (2021), are also not affected by clouds, but the position of the instrument's field of view (FoV) at the OH* airglow layer height depends strongly on the wind at the balloon height.





The natural phenomenon of the OH* airglow is due to different rotational-vibrational transitions of the excited hydroxyl molecule emitting radiation between ca. 0.6 and 4.5 μm. Table 1.1 of Khomich et al. (2008) provides an overview of the principal airglow emissions, including OH*. Additionally, the Skycorr model described in Noll et al. (2014) and available at https://www.eso.org/observing/etc/bin/gen/form?INS.MODE=swspectr+INS.NAME=SKYCALC can be used to show the airglow spectrum in an individual spectral range[1]. The brightest OH* airglow transitions appear in the near infrared around

1500 nm (Rousselot et al., 2000). Some of the transitions with low rotational quantum numbers are approximately in local thermodynamic equilibrium. From those, kinetic temperature can be estimated. Even if kinetic temperature cannot be derived, the OH* airglow can be used as dynamical tracer.

The OH* airglow forms a layer in the upper mesosphere / lower thermosphere (UMLT). Often, height and full width at half maximum (FWHM) are given as 87 km and 8 km, respectively, referring to Baker and Stair (1988). Those authors summed

up results from 34 rocket flights at mid- and high-latitudes, which were conducted by different experimenters. They derived $86.8 \pm 2.6$ km for the altitude of the peak and $8.6 \pm 3.1$ km for the thickness of the OH* airglow layer. Both parameters can vary (in the case of the centroid height by some kilometres) over several days or even during a single night due to strong dynamics (e.g., changes in the residual circulation during a stratospheric warming or strong tidal motions). Peak and centroid height differ slightly since the OH* airglow profile is not completely symmetric: the layer fraction above the peak height is

larger than below (Moreels et al., 1977) and the emission rate profile can be modelled by an asymmetric Gaussian distribution with a wider top part and a narrower bottom part (Khomich et al., 2008). However, this asymmetry is not significant for most applications and if derived from satellite limb measurements in most cases smeared out at least in part due to the size of the instrument's FoV. Long-term satellite-based investigation of a larger geographical region (52°N - 52°S, Wüst et al. (2020) and references therein) confirm the mean height values of Baker and Stair (1988) although there is some geographical variation.

Amongst others, the height of the emission peak depends on the exact vibrational level: adjacent vibrational levels are separated by some 100 m (Baker and Stair, 1988; Adler-Golden, 1997; Von Savigny et al., 2012). If one refers to different vibrational transitions, one therefore obtains information weighted toward different heights. In contrast, a zenith-viewing instrument which spectrally integrates over different vibrational transitions, receives its signal from a greater height range than one which observes an individual vibrational transition.

---

[1] This model was developed for astronomers. In order to use it for the derivation of OH airglow information remove the tick at "Optional parameter initialisation using Almanac Service" and select "Emission lines of the upper atmosphere" as well as "Airglow/residual continuum". The model includes the van-Rhijn effect (through "Altitude of target above horizon" or "Airmass") and some variations with month and night. Additionally, you can include the influence of the sun and of precipitable water vapor on the airglow lines (which is especially important at ca. 1.4 and 1.9 μm, see also Smette, A., Sana, H., Noll, S., Horst, H., Kausch, W., Kimeswenger, S., Barden, M., Szyszka, C., Jones, A., and Gallenne, A.: Molecfit: A general tool for telluric absorption correction-I. Method and application to ESO instruments, Astronomy & Astrophysics, 576, A77, 2015.).



OH* airglow measurements are often referred to as mesopause measurements. This is strictly speaking only valid during
summer since the mesopause changes in height by ca. 14 km during the year: it reaches ca. $86 \pm 3$ km in summer (approximately
May–August in the northern hemisphere according to She et al. (2000)) and ca. $100 \pm 3$ km in winter (rest of the year). Based
on lidar measurements for different latitudes (Lübken and Von Zahn (1991), She et al. (1993), Von Zahn et al. (1996)) and
models (Berger and Von Zahn (1999)) this mesopause height variation is a worldwide phenomenon. However, the OH*

centroid height varies only by a few kilometres, so OH* airglow measurements sample the UMLT in different seasons. That
means the OH* airglow layer lies in a height region with a small vertical temperature gradient during summer and with negative
vertical temperature gradient during winter. This has direct consequences, for example, for the estimation of the mesopause
temperature trend or of the Brunt-Väisälä frequency based on OH* airglow temperature measurements.

Ground-based and airborne measurements integrate over the OH* airglow layer profile. Thus, vertically resolved information

is not available or only to a very limited extent (as already mentioned above using different vibrational levels). Satellite-based
limb scanning instruments can provide vertically resolved profiles of the OH* volume emission rate (VER). There exists a
large number of different ground-based instruments used for OH* airglow observation all over the world. Several stations are
loosely working collaboratively in the coordinated Network for the Detection of Mesospheric Change (NDMC,
https://ndmc.dlr.de). The largest sub group of technical identical OH* spectrometers in the NDMC are of the GRIPS (Ground

based Infrared P-branch Spectrometer) type. These instruments were developed at the University of Wuppertal in the late
1970s. In the 2000s, the instruments were re-designed by the German Aerospace Centre (DLR) and the University of Augsburg
(UNA) to take advantage of the technical developments in semi-conductor technology. This has allowed operation without
liquid nitrogen cooling and an order of magnitude improvement in temporal resolution. As for most instruments in the present
era, the retrieval algorithm was designed and standardized to facilitate automated operation. This has enabled the simultaneous

operation of several instruments as a group to view a contiguous atmospheric region, such as the Alpine region where currently
seven GRIPS instruments are deployed and contribute to the Virtual Alpine Observatory (VAO, https://www.vao.bayern.de).
SATI (Spectral Airglow Temperature Imager) is another type of OH instruments, in this case imagers, which are deployed at
several sites (Sargoytchev et al., 2004). It is a complete redesign of an older instrument called MORTI, which was not sensitive
to OH (Wiens et al., 1997). Beside adding the OH channel, the cryogenic cooling was replaced by thermo-electric one,

temperature and emission rate are readout in real time, and it can be operated remotely. SATI is a Fabry–Perot interferometer.

As for the satellite-based limb scanning instruments, there are a few of them currently in orbit and providing measurements
for many years: SABER (Sounding of the Atmosphere using Broadband Emission Radiometry) on TIMED (Thermosphere
Ionosphere Mesosphere Energetics Dynamics), OSIRIS (Optical Spectrograph and InfraRed Imager System) on Odin and
MLS (Microwave Limb Sounder) on EOS (Earth Observing System) Aura (which is somehow extraordinary since it measures

the lowest possible rotational transition of OH, OH is not vibrationally excited), for example.

This report provides an overview of results focussing on dynamics derived from long-term OH* airglow measurements, but
also includes measurements from a larger spatial domain, i.e., using multiple instruments, in section 4. This overview and



today's technical and computational development is the basis for identifying possible questions of future research outlined in section 5. Before doing that, the main points leading to the formation of the OH* airglow layer and the response of the OH* airglow layer to dynamical disturbances are summarised in section 2. The principle of the temperature derivation is outlined in section 3.



## 2 The OH* layer

### 2.1 An overview of OH* production and loss mechanisms in the UMLT

Excited OH molecules are produced by different reactions in the UMLT. A comprehensive overview is given by Khomich et
al. (2008) in their chapter 2.2.

The first two reactions were identified in the early 1950s: the **ozone-hydrogen reaction** (Bates and Nicolet (1950), Herzberg
(1951), Heaps and Herzberg (1952))

$$O_3 + H \rightarrow OH^*(v \leq 9) + O_2 + 3.34 \, eV \tag{1}$$

and the **oxygen-hydrogen reaction** (Krassovsky, 1951)

$$O_2^* + H \rightarrow OH^* + O \tag{2}$$

were followed by the **perhydroxyl-oxygen reaction** (Krassovsky (1963), Nicovich and Wine (1987), Nicolet (1989))

$$O + HO_2 \rightarrow OH^*(v \leq 6) + O_2 \tag{3}$$

in the 1960s.

Vibrationally excited OH molecules (vibrational level $v \leq 9$) emit in the spectral range of ca. 0.6–4.5 µm. In general,
vibrational transitions are accompanied by rotational transitions (for example, Andrews, 2000), which are less energetic. The
rotational transitions split into the P-, Q-, and R-branch depending on the change of the quantum number of the total angular
momentum J. The P-branch is the branch with lowest energy and the R-branch the one with the highest, the Q-branch is in
between. So, for emission, as it is the case for the topic of this publication, ΔJ increases through the rotational transition by
one for the P-branch, decreases by one for the R-branch, and stays constant for the Q-branch. In the following paragraph, the
focus is on the vibrational transitions; rotational transitions will be addressed again in section 3, in the context of rotational
temperature derivation.

In general, the ozone hydrogen reaction is regarded as the most important one for the generation of OH* (Brasseur and Solomon
(1986), Grygalashvyly (2015) and citations therein) reaching vibrational levels up to 9. Lower levels are populated in a
radiative cascade by spontaneous emission

$$OH^*(v') \rightarrow OH^*(v'') + h\nu \tag{4}$$

with $v' > v''$.

Since only the ozone-hydrogen reaction is able to populate the higher vibrational states (e.g., $v'>6$), modelling OH* intensity
can be simplified by focusing, for example, on the OH(8-3) band, leaving out the perhydroxyl-oxygen reaction, rather than on
the OH(4-2) or OH(3-1) bands, which are frequently measured.






From the ozone hydrogen reaction, it becomes clear that the OH concentration strongly depends on the availability of $O_3$ and H.

$O_3$ is produced by a three-body-collision reaction of O and $O_2$ as well as a collision partner which facilitates the reaction but remains unchanged afterwards. Due to the composition of the Earth atmosphere the collision partner is $N_2$ in most cases. The
number density of $N_2$ and $O_2$ decreases exponentially with height in the homosphere, however, atomic oxygen shows a different behaviour (see Figure 1 a and b, left panel, for the distinction between day and night and a greater height range). It is generated by the dissociation of $O_2$ and follows a Chapman layer very well above 80 km height (e.g., Swenson and Gardner (1998), for consideration of the altitude range of up to 220 km see Rodrigo et al. (1991)). A Chapman layer is driven by the availability of radiation, which is high above the peak of the Chapman layer and decreases with decreasing height due to absorption. This
radiation interacts with an absorber, in this case $O_2$ ,whose density is high below the peak and decreases with greater heights (see, for example, Andrews (2000) for the explanation of a Chapman layer). The peak of this atomic oxygen layer is above 90 km height and shows a strong gradient below. The availability of the different reaction partners leads to the formation of an $O_3$ layer in the UMLT.

The number density of H reaches its maximum between 80 and 100 km height (Figure 1 a and b, left panel). Below ca. 90–
95 km height, the number densities of H and $O_3$ fluctuate around each other. The relative variation of $O_3$ and H shown in Figure 1 (a and b, left panel) illustrates why OH* forms a layer in the UMLT. This layer is approximately Gaussian (see e.g. figure 1 in Swenson and Gardner (1998)).

The produced OH* has a very short lifetime, it takes less than 1 s before it relaxes through collision (so in this case the excess energy is transferred to heat) or through spontaneous emission of infrared radiation via equation 4 (Swenson and Gardner,
1998). Additionally, it can deactivate by chemical reaction with atomic oxygen:

$$OH^* + O \rightarrow O_2 + H \tag{5}$$

This process mainly occurs on the top side of the layer, where the availability of atomic oxygen is higher than on the bottom side.

Due to increasing air density with decreasing height, collisional quenching with molecules of the surrounding atmosphere
becomes more probable on the bottom side and dominates the loss term there (the surrounding atmosphere is in the following denoted by $M$ which typically is molecular nitrogen, or molecular or atomic oxygen), i.e.

$$OH^*(v') + M \rightarrow OH^*(v < v') + M \tag{6}$$

Quenching by atomic oxygen (Grygalashvyly, 2015), may be supplemented by multi-quantum collisional deactivation with molecular oxygen (Von Savigny et al. (2012) and references therein) and is the reason why emission peak altitudes of OH
emissions from adjacent vibrational levels differ on average by ca. 400–700 m (Baker and Stair, 1988; Adler-Golden, 1997; Von Savigny et al., 2012; Noll et al., 2016).





(a)

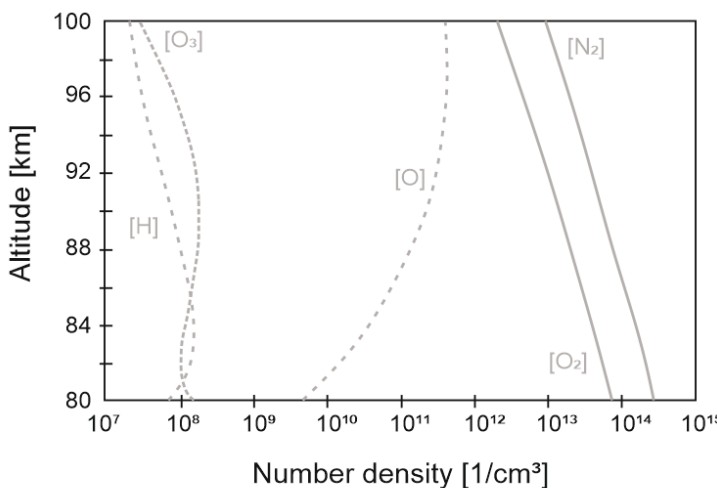

(b)

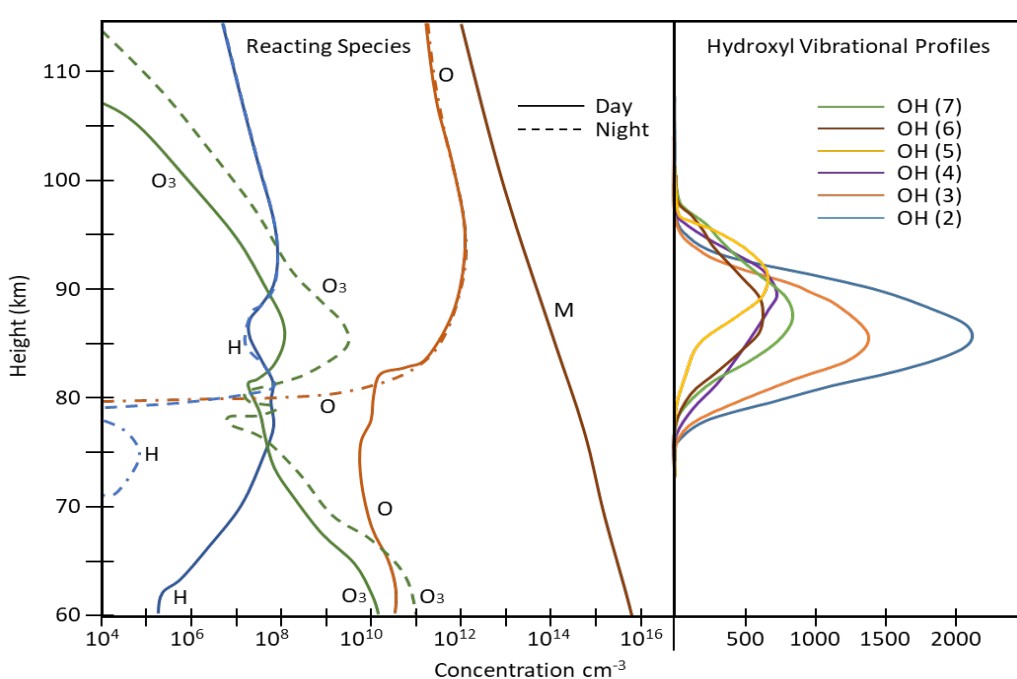

**Figure 1 a) Number density of the two major (O₂ and N₂, solid lines) and the three minor species (O₃, O and H, dashed lines) involved in the production of OH\*  between 80 and 100 km height following Swenson and Gardner (1998) according to MSIS-90. Relatively large uncertainties exist for the number density of atomic oxygen (Richter et al., 2021), which also shows seasonal variations (see, e.g., figure 2.15 in Khomich et al. (2008)). b) Left panel: Vertical profiles of the major reacting species in the OH photochemical**



**scheme for daytime (solid) and nighttime (dashed) conditions between 60 and 115 km height, M denotes a collisional partner (adapted from Battaner and Lopez-Moreno (1979)). Right panel: OH profiles for $v' = 2–7$ (adapted from Lopez-Moreno et al. (1987)).**

This was a qualitative explanation for the existence of the OH* layer in the UMLT. Grygalashvyly (2015), for example,
introduces several analytical approaches to OH* layer parameters.

Until this point, neither dynamical processes which transport species needed for the OH chemistry nor possible variations in the availability of the different constituents due to solar radiation are considered (e.g., during one day, see Lowe et al. (1996) for the behaviour of the OH layer post sunset). Since the focus of this paper is on dynamical processes, only the first aspect is addressed in the following section.

A good overview on models of wave interactions with OH* chemistry is given by Snively et al. (2010) in their section 1.3.

## 2.2 Influence of atmospheric dynamics on the OH* airglow layer characteristics

The UMLT is dynamically driven over the year by the residual circulation, which is due to an interaction between gravity waves and the mean flow, and on temporal scales of weeks to minutes by different atmospheric waves (planetary waves, tides,
gravity waves, and acoustic waves). All these processes influence advection, modifying the density of the respective species, as well as the temperature that changes the chemical reaction rates. Each of the dynamical processes does this on different temporal scales.

The reacting species for the ozone-hydrogen reaction, $O_3$ and H, but also OH* itself have relatively short life times in the UMLT but they still differ by an order of magnitude (ca. 4-6 min for $O_3$ and H, and less than 1 s for OH*, e.g., Snively et al.
(2010) and Swenson and Gardner (1998)). That means that OH* decays before it can be transported substantially. Concerning $O_3$ and H, transport depends on the period of the disturbance: if it is in the range of the life time of $O_3$ and H (such as gravity waves with frequencies in the vicinity of the Brunt-Väisälä frequency or acoustic waves), advection needs to be taken into account (see, e.g., Inchin et al. (2020) who also use the chemical model of Snively et al. (2010) for acoustic waves).

$O_3$ is arises from O and $O_2$. Both of their lifetimes are considerably different from those of H and $O_3$: for O, the lifetime
exceeds one day at the UMLT (Brasseur and Solomon, 1986). Since there is less O available than $O_2$ between 80 and 100 km (Figure 1 a) and O has a steeper vertical gradient than $O_2$, O controls the formation of $O_3$. Swenson and Gardner (1998) show in their figure 7a the individual contributions of O, $O_2$ and the temperature to the vertically-resolved volume emission rate of OH(8-3). So, dynamical processes leading to vertical transport within a day or less can change the O concentration with height and influence the formation of $O_3$, and in turn the generation of OH*. The downward transport of O leads to a lower centroid
height and to a brighter and thicker OH* airglow layer (e.g., Liu and Shepherd (2006)).





Concerning the vertical transport in the UMLT, the residual circulation plays a prominent role. Accordingly, analyses of the OH* airglow layer height reveal an annual cycle. It is more pronounced at higher latitudes than at lower ones (Wüst et al., 2020). A semi-annual cycle is also often found. It is either attributed to physical mechanisms or treated as an analysis artefact, which is necessary to compensate for the fact that the annual cycle does not correspond exactly to a sine wave. Grygalashvyly

(2015) provided a good overview about the different characteristics together with explanation of their origins in his section 3. Becker et al. (2020) estimated the effect of gravity waves with horizontal wavelength up to 1000 km on height and number density of the OH* airglow layer using model studies.

Due to the strong vertical gradient of O, the maximum perturbation of the OH(8-3) volume emission rate is ca. 2–4 km below

the peak of the OH* layer (measured, for example, by Lowe et al. (1996) and calculated, for example, by Swenson and Gardner (1998)). The height of the maximum rotational temperature perturbation, however, can be found relatively close to the peak height of the OH* layer. That means if the OH* intensity is measured and rotational temperature is derived from these measurements, both signals originate from different altitudes. This leads to a phase shift for dynamical features derived from temperature and intensity time series.

Rotational temperature and intensity derived by a ground-based instrument looking into the sky are always integrated and therefore averaged over the OH* layer profile. Swenson and Gardner (1998) showed that perturbations of this rotational temperature and of the measured relative intensity differ only by a factor, which they called $g_T$ in the description of the rotational temperature perturbation and $g_{OH}$ in the case of relative intensity. The first one ($g_T$) depends amongst others on the temperature profile (which varies during the year), while this is not the case for $g_{OH}$ which is determined by parameters of the

OH* layer only. These factors can be interpreted as height-dependent weights, which significantly influence the averaging of the respective signal over the OH* layer. Their different variation with height (see Figure 2) is the reason that the observed relative amplitude of a disturbance depends on whether one investigates rotational temperature or intensity. For gravity waves with long vertical wavelengths (> ca. 20 km, or high horizontal phase speed > ca. 65 m/s), the relative intensity perturbation is larger while the relative temperature perturbation is smaller than the true atmospheric density perturbation. Furthermore,

time series of intensity measurements are more suitable than time series of rotational temperatures time series for the study of shorter vertical wavelengths (or slower gravity waves).

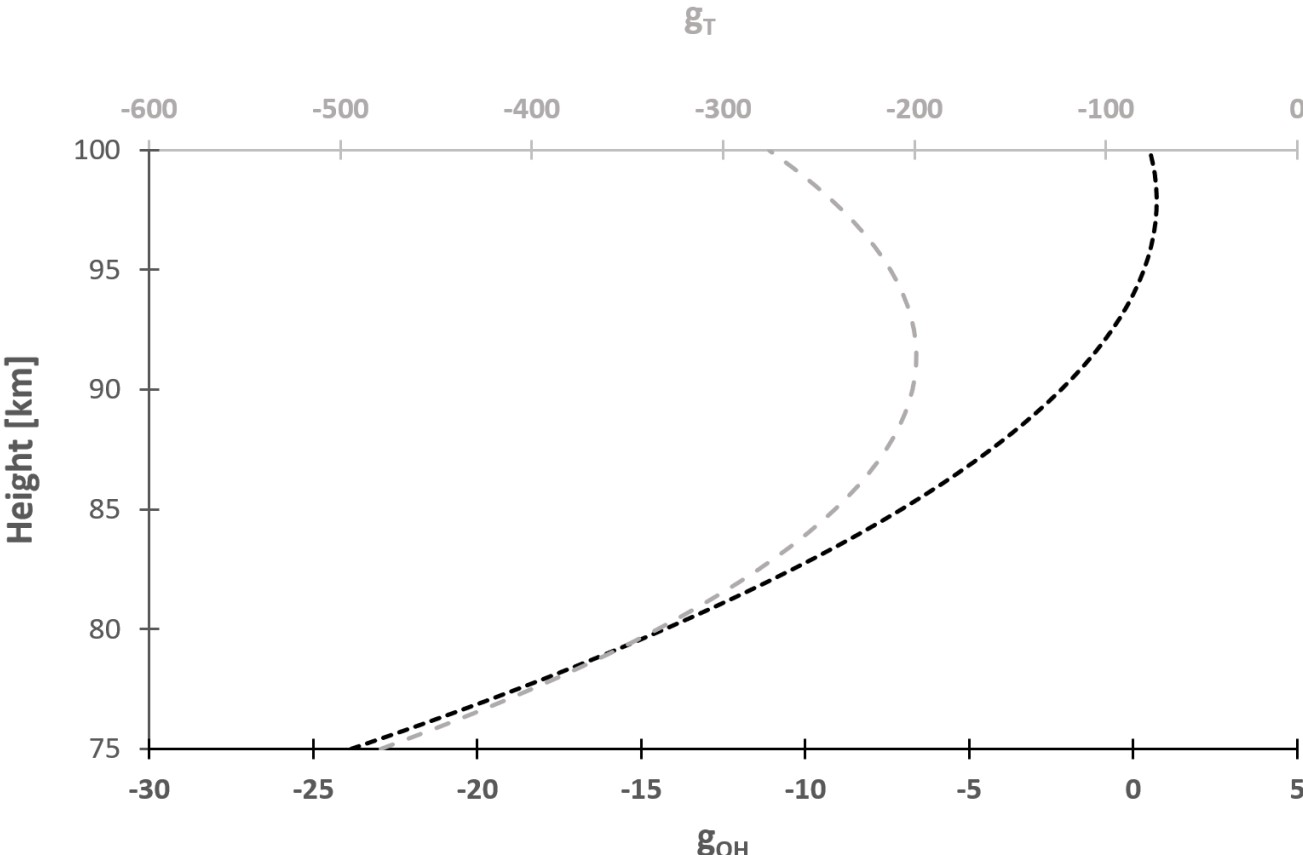

**Figure 2 Height-dependent development of $g_T$ (grey) and $g_{OH}$(black) according to the formulas (42) and (29) of Swenson and Gardner (1998) calculated for January: while $g_{OH}$ decays nearly linearly to zero, this is not the case for $g_T$. Among other arithmetic operations, these factors are multiplied by the relative density perturbation and the undisturbed VER in order to derive the rotational temperature and the relative intensity perturbation. So, it becomes clear that relative intensity perturbations suffer less from averaging effects than the rotational temperature perturbation.**



## 3 Measurement techniques for OH* intensity and derivation of temperature

The first measurements of the OH* airglow were performed in the visible range by ground-based instruments as early as 1920/30s by Babcock, Slipher, and Sommer as Meinel (1950a) mentions. However, an identification of the peaks was not possible during that time due to the poor spectral resolution of the spectrogram measurement. In the following years, the observation techniques improved and the existence of OH molecules in the atmosphere was discussed (for an overview see chapter 2.2.1 of Khomich et al. (2008)). In 1949/50, Meinel operated a spectrograph based on photo plates at Yerkes Observatory (42° 34′ 13″ N, 88° 33′ 23″ W) which he exposed for four hours to the night sky. He experimentally confirmed the spectral lines to be rotational-vibrational transitions of the OH* molecule (Meinel, 1950a) and calculated a temperature from them (Meinel, 1950b).

In the vast majority of cases, ground-based instruments deliver vertically integrated values of the OH* airglow volume emission rate. If the instruments have the required spectral resolution and local thermodynamic equilibrium (LTE) conditions in the emitting region are fulfilled, information about the kinetic temperature, which is naturally weighted by the OH* VER, can be derived. The first criterion is a technical requirement, whereas the second one depends on the individual OH* rotational vibrational transition.

Generally speaking, the assumption of LTE is valid for vibrational and rotational modes below ca. 60 km height (Andrews, 2000). Above this height, the relation of the times needed for spontaneous relaxation of a transition and for relaxation due to collision need to be investigated in more detail: for LTE, the mean time between two collisions has to be much shorter than the lifetime for radiative decay. In general, the lowest rotational transitions of the lower OH* vibrational transitions are sufficiently close to LTE for a reliable estimate of the kinetic temperature (see, e.g., Sivjee (1992), Pendleton Jr et al. (1993) and references therein or Noll et al. (2020) who provide in their figure 13 an estimate of non-LTE contributions to some rotational temperatures based on their own work and the one of Oliva et al. (2015)).

The temperature $T_{rot}$ can be calculated from two different lines (Meinel, 1950b; Baker, 1978; Baker et al., 1985). The basis of this calculation is the following relation for the volume emission rate $I$ (in photons s$^{-1}$ cm$^{-3}$) for spontaneous emission:

$$I_{(v',J' \rightarrow v'',J'')} = N_{v',J'} A_{(v',J' \rightarrow v'',J'')} \tag{7}$$

where $v$ and $J$ are the vibrational and rotational levels, primed values denote the initial (so upper) state, two primed values the final (so lower) state. $N_{v',J'}$ is the number density of molecules (in cm$^{-3}$) in the initial state which undergoes the transition to the end state. $A_{(v',J' \rightarrow v'',J'')}$ is the Einstein coefficient of this specific spontaneous emission (in s$^{-1}$), which describes the transition probability. If LTE holds for the rotational transitions, the distribution of the molecules over the different rotational quantum numbers follows the Boltzmann distribution and

$$N_{v',J'} = \frac{N_{v'}}{Q_{v'}(T)} (2J' + 1) \exp\left(-\frac{E_{v',J'}}{k_B T_{rot}}\right) \tag{8}$$





$N_{v'}$ is the total number of molecules in the vibrational level $v'$ and $Q_{v'}(T)$ is the partition function or state sum defined (Thorne, 1988):

$$Q_{v'}(T) = \sum_{J'}(2J' + 1)\exp^{\left(-\frac{E_{v',J'}}{kT_{rot}}\right)} \tag{9}$$

which is a function of temperature.

The factor $(2J' + 1)$ is due to the degeneracy of the state. $E_{v',J'}$ is the upper state rotational energy, $k_B$ the Boltzmann constant

and $T_{rot}$ the rotational temperature.

Substitution of $N_{v',J'}$ in (7) according to equation (8) gives:

$$I_{(v',J' \to v'',J'')} = \frac{N_{v'}}{Q_{v'}}(2J' + 1)\exp\left(-\frac{E_{v',J'}}{k_B T}\right)A_{(v',J' \to v'',J'')} \tag{10}$$

So, the intensity of the lines of a rotational-vibrational transition depends non-linearly on the rotational temperature. Equation

() can be re-formulated as follows:

$$\underbrace{ln\left(\frac{I_{(v',J' \to v'',J'')}}{A_{(v',J' \to v'',J'')}\cdot(2J'+1)}\right) - ln\left(\frac{N_{v'}}{Q_{v'}}\right)}_{=:b} - \underbrace{\frac{1}{k_B T_{rot}}}_{=:m}\cdot E_{v',J'} = 0 \tag{11}$$

Equation (11) now describes a linear relation, so a straight line with $b$ as the y-axis intercept and the line slope $m$. The term $\frac{I_{(v',J' \to v'',J'')}}{A_{(v',J' \to v'',J'')}\cdot(2J'+1)}$ is partly known—even though there is some discussion about the true value of Einstein coefficient (see

Noll et al. (2020) and references therein and tables 2.4 – 2.13 in Khomich et al. (2008) for an overview of some of them)—and can partly be taken directly from the airglow measurement. $\frac{N_{v'}}{Q_{v'}}$ is not known but constant and $E_{v',J'}$ is known. Measuring the OH* airglow intensity for at least two different rotational transitions of one vibrational branch and plotting the intensity versus $E_{v',J'}$ allows fitting a line with a slope determined by the inverse of the rotational temperature and the Boltzmann constant. The advanced mesospheric temperature mapper (Pautet et al. (2014) and references therein) is an example of an

instrument where only a pair of lines ($P_1(2)$ and $P_1(4)$ of OH(3-1) band) are used. In many cases, three or more rotational lines are used, which makes the fit more robust, and provides an immediate measure of the uncertainty of the temperature retrieved (e.g., French et al. (2000), Sigernes et al. (2003)).

Vibrational transitions with $\Delta v = 1$ are the brightest (see Figure 4.6 from Roach and Gordon (1973) or Krassovsky et al. (1962) page 900). From the ground, only bands with $\Delta v \geq 2$ can be detected, which means that the emitted wavelength is shorter or



equal to 2.1 µm (Khomich et al., 2008). Ground-based OH* airglow spectrometers in many cases use the OH(3-1) or OH(4-2) transitions in the wavelength region 1.5-1.6 µm where atmospheric absorption is minimised (see e.g., Bellisario et al. (2020) also Chadney et al. (2017), and Espy and Hammond (1995)). The P-branch of these bands is preferred since the energy difference between the different rotational transitions increases with increasing rotational quantum number. Therefore, the free spectral range is larger than for the Q- and R-branches.


Once the temperature has been determined from a number of lines within a band, the intercept in (11) allows a value of $ln\left(\frac{N_{v'}}{Q_{v'}}\right)$ to be calculated. A knowledge of the behaviour of $Q_{v'}(T)$ as a function of temperature (Krassovsky et al., 1962) then provides a method of calculating $N_{v'}$, the number of OH* in the upper vibration state of the band under study. The total intensity of the band can then be determined (Hecht et al., 1987) from

$$I_{v' \to v''} = N_{v'} A_{v' \to v''} \tag{12}$$

where $A_{v' \to v''}$ is the Boltzmann-averaged Einstein coefficient. This latter value is independent of temperature over the range of values obtained in the altitude range of the OH* layer.

While the method of calculating band intensity values just described is straightforward, the majority of intensity measurements reported are relative values however. For a given instrument, relative intensity values are suitable for many studies, and it

avoids the need to address the difficult twin problems of intensity calibration of instruments and taking account of atmospheric extinction. Unfortunately, the use of relative intensity values makes it impossible to compare results from different instruments. This makes the relatively few reports of intensity values (e.g., Harrison and Kendall (1973), Sivjee and Hamwey (1987), Hecht et al. (1987), Mulligan et al. (1995), Espy and Stegman (2002)), all the more valuable. Comparison of result from ground-based instruments with VER measurements from satellite instruments such as SABER on TIMED provides a

way for ground-based observers to obtain approximate values for their intensity values, and thereby compare their results with other observers.

Since the first observation of OH* airglow, the technique improved according to the developments in instrumentation detector materials (higher spectral resolution and/or increased signal-to-noise ratio, SNR), and in digitalization, but also the determination of parameters needed for the temperature retrieval (e.g., Einstein coefficients) have evolved since then. Photo

plates were replaced by single cell Germanium (Ge) detectors (Shemansky and Jones, 1961) and then by further materials such as Gallium Arsenide (GaAs) or Indium Gallium Arsenide (InGaAs) (see e.g., Misawa and Takeuchi (1978, 1977): OH(8-3) around 727 nm, InGa photomultiplier; Scheer (1987): airglow between 845–867 nm, GaAs photomultiplier, Mulligan et al. (1995): airglow at 1–1.65µm, one cell InGaAs detector, Greet et al. (1998): airglow 800–900 nm, GaAs photomultiplier; Suzuki et al. (2008): airglow at 800–1700 nm, InGaAs detector array; Schmidt et al. (2013): airglow at 1.5–1.6 µm, InGaAs

detector array). The SNR is worse for InGaAs compared to the Ge detector operated in the 1980s and 1990s, e.g., by Bittner





et al. (2000), however, since the noise is relatively constant with time and regularly determined during a dark current measurement, this disadvantage could be overcome. While the Ge detector used by Bittner et al. (2000) needed liquid nitrogen cooling for a good SNR ratio, InGaAs detectors deliver spectra at higher detector temperatures, which can be achieved by thermoelectric cooling, which is much more practical than the use of liquid nitrogen. The elimination of liquid nitrogen cooling

facilitates unattended operation. Line detectors or linear detector arrays (Schmidt et al., 2013) allow the simultaneous measurement of a larger spectral range and avoid scanning mechanisms. This improves the temporal resolution of OH* airglow spectrometer measurements by an order of magnitude, from some minutes to ca. 10 seconds, and enables at least in parts to address the infrasound spectrum (Pilger et al., 2013). The automation of the temperature retrieval from the spectra allows the rapid analysis of spectra.

A new development published by the University of Wuppertal is the spatial heterodyne interferometer GRIPS-HI, which is a hybrid combining properties of both, grating spectrometers and Fourier interferometers (Stehr et al., 2021). It is sensitive to 1520–1550 nm. Its advantage over grating spectrometers is the spectral resolution and the optical throughput, which are regarded as main sources of measurement uncertainty. Furthermore, temperature measurements can be resolved spatially in one dimension even when the instrument is not scanning (Harlander et al., 2002). However, until now only a prototype of

GRIPS-HI exists, which still needs some improvement.

When using an airglow spectrometer based on a single detector cells or a line detector, the OH* airglow signal is averaged horizontally over the FoV. The size of the FoV depends on the optics of the spectrometer but also on the zenith angle at which the instrument is operated. This needs to be taken into account when interpreting the results especially with respect to gravity wave parameters (in the context of satellite validation the reduced sensitivity is known as observational filter). The influence

of different zenith angles on the sensitivity of an OH* airglow spectrometer for gravity waves is shown in Wüst et al. (2016), for example. With airglow spectrometers based on a single detector cells or a line detector horizontal wave parameters cannot be derived straight away—camera systems would be required for this.

Different to OH* airglow spectrometers, the values of OH* cameras are not averaged over the whole FoV, their spatial resolution depends on the number of pixels on the chip, the objective used and the zenith angle of the instrument. Effects of

spatial resolution need to be considered when results of different instruments are compared. When using OH* airglow cameras with suitable filter wheels which are sensitive to different rotational transitions of one vibrational band, temperature information can be retrieved. The only publication reporting a temperature imaging system, so not a scanning one, is by Pautet et al. (2014). Technical developments have made the imaging systems smaller and the instruments more compact, so that they are relatively easy to deploy on airplanes, for example. The history of imaging wave structures in the OH layer until the late

1990s is reviewed in Taylor (1997).





## 4 Results from OH* airglow measurements

A large number of publications are available on the topic of analysing OH* airglow data measured from ground, starting with
Krassovsky (1972). The FoV and spatial and temporal resolutions of an instrument limit the range of dynamic phenomena that
can be observed. For example, the temporal resolution of most instruments is not well-suited for infrasound observations, and
in the case of imaging systems, the spatial resolution does not allow the display of planetary waves. However, planetary waves
can be addressed in the temporal domain by spectrometers and cameras, and therefore individual OH* airglow measurements
allow the observation of travelling planetary waves or show the superposition of stationary and travelling planetary waves.
Generally, it is not necessary to derive the rotational temperature for the analysis of gravity waves based on OH*
measurements. As mentioned earlier, wave-induced fluctuations in intensity are larger than in temperature and smaller vertical
wavelengths can be derived from intensity time series compared to rotational temperature time series. However, for the
derivation of some gravity wave parameters, knowledge about the wave-induced temperature perturbation is needed.

The advantage of ground-based OH* airglow observations is that some of them have been made over many years. This allows
the investigation of the long-term temperature development or long-term analyses of gravity wave parameters, for example.
The results of some publications with the longest time series focussing on gravity waves (or dynamics) are summarized in the
following.

Concerning airborne results, the number of publications is limited as OH* airglow instruments are not a suitable equipment
for commercial aircraft (such as CARIBIC for the troposphere). They need special ports in the upper part of the aircraft
through which they can receive incoming light and which incurs high-cost and licence issues amongst others. Therefore, they
are only operated on research aircraft on a campaign basis. Among the campaigns reported in the literature are, e.g., EXL98,
Leonid MACC, ALOHA-90 and -93, DEEPWAVE, and GW-LCYCLE, which are addressed in the following.

Analyses of OH* airglow spectroscopy time series directly deliver gravity wave periods and their amplitudes in intensity and
temperature. The respective parameters are subject to vertical, horizontal and temporal averaging. For most scientific
questions, the temporal averaging is negligible (the temporal resolution is mostly in the range of seconds to minutes). OH*
airglow imagery additionally allows the extraction of horizontal wavelengths of gravity waves, of their horizontal phase
velocities, and of their horizontal propagation direction. Spatial scanning spectrometers are able to do the same (Wachter et
al. (2015); Wüst et al. (2018)), however, their sensitivity to horizontal structures is limited since a scanning spectrometer is
always a compromise between the time needed for the scan, the size of the scanned region, and the size of the individual FoV.

Based on the dispersion relation for gravity waves or its simplifications according to the observed period range, further wave
parameters such as the vertical wavelength of gravity waves can be calculated (e.g., Swenson and Liu (1998); Suzuki et al.
(2007) for imaging systems; Wüst et al. (2018) for a scanning spectrometer):





$$m^2 = \frac{(k^2+l^2)(N^2+\hat{\omega}^2)}{\hat{\omega}^2-f^2} - \frac{1}{4H^2} \qquad (13)$$

with     $k, l, m$ the zonal, meridional, and vertical wave number,

$N$        the Brunt-Väisälä frequency,

$f$        the Coriolis frequency,

380        $\hat{\omega}$        the intrinsic GW frequency ($\hat{\omega} = \omega - u \cdot k - v \cdot l$ with $u, v$ the zonal and meridional background wind and $\omega$ the GW frequency), and

$H$        the scale height.

However, for the application of the dispersion relation additional information such as the Brunt-Väisälä frequency and wind velocities is mandatory.

Based on OH* airglow transitions, horizontal wind velocities can be measured using Fabry-Perot interferometers (e.g., Shiokawa et al. (2001)). However, as far as we know, the data do not allow the derivation of gravity wave induced fluctuations due to relatively large uncertainties, which limits its use for the calculation of kinetic GW energy, for example. A possibility with a better time resolution and lower uncertainties are Michelson interferometers such as ERWIN (Gault et al., 1996) and ERWIN II (Kristoffersen et al., 2013). They also allow two-dimensional imaging of line-of-sight winds (Langille et al., 2013).
Alternative non-airglow wind measurement techniques such as lidar or radar achieve a temporal resolution in the range of an hour or less (e.g., Rauthe et al. (2006), Jacobi et al. (2009)).

The Brunt-Väisälä frequency cannot be taken from OH airglow measurements, it can at best be estimated especially during summer when the OH* layer is in the range of a small vertical temperature gradient. So, coincident lidar (Tang et al., 2002) or satellite measurements of the vertical temperature profile or climatologies of the Brunt-Väisälä frequency (e.g., Wüst et al.
(2017); Wüst et al. (2020)) are necessary. Then, the density of gravity wave potential energy (GWPED) can be calculated, when the rotational temperature measurements are separated into wave-induced temperature variations and background temperature. Based on the amount of potential energy, kinetic energy can at least be estimated: from linear gravity wave theory, the ratio of kinetic (mostly horizontal) to potential energy is predicted to be equal to ca. 5/3 (Van Zandt, 1985). In order to quantify gravity wave activity without providing kinetic or potential energy (density), the variability of the data in
terms of standard deviation or based on spectral analyses is provided by some authors (e.g., Offermann et al. (2009), López-González et al. (2020) and citations therein). Since the majority of imaging systems are not able to derive temperature, GWPED is in most cases only calculated from spectrometer measurements.





An alternative way to derive the vertical wavelength of GW without information about the horizontal wind is the measurement of different vibrational transitions (Schmidt et al., 2018) since their respective centroid height differs by ca. 500 m for adjacent upper vibrational levels (see Figure 1 b, right panel). Furthermore, the fact that the signal of maximal temperature variations originates from another height than the signal of maximal intensity variation (the height of the maximum intensity perturbation is lower than the height of the maximum temperature perturbation) can be used (Reisin and Scheer, 2001). This leads to a phase shift $\phi$ when the wave is derived from rotational temperature and intensity time series and this phase shift is related to the vertical wavelength $\lambda_z$ by

$$\phi = \frac{I_s}{\lambda_z} 360° \qquad (14)$$

with $I_s$ the effective layer separation (which was given by ca. 3.75 km according to Swenson and Gardner (1998) for their model setup). Since the effective layer separation can vary, this method gives at least an estimate for the vertical wavelength. As already mentioned earlier, the sensitivity of intensity and temperature to small vertical wavelengths is limited by the integration of the variation through the OH layer profile (typically ~8km). This limits the application of equation (14).

If the vertical and horizontal wavelengths, $\lambda_z$ and $\lambda_h$, are known, momentum and energy flux, $F_M$ and $F_E$, can be estimated from the volume integrated OH* intensity based on the so-called cancellation factor $CF$, which is determined by the ratio of the relative intensity and the relative atmospheric temperature (relative in the sense of wave-disturbed value divided by undisturbed value, Swenson and Liu (1998), Suzuki et al. (2007)) with

$$F_M = \frac{6 \cdot 10^4 \cdot \lambda_z \cdot (I'_{OH})^2}{\lambda_h \cdot CF^2 \cdot (I_{OH})^2} \qquad (15)$$

and

$$F_E = \frac{2.3 \cdot 10^{-3} \cdot \lambda_z{}^2 \cdot (I'_{OH})^2}{\lambda_h \cdot CF^2 \cdot (I_{OH})^2} \qquad (16)$$

where $I'_{OH}$ is the wave-perturbed volume integrated OH* intensity and $I_{OH}$ the unperturbed one.

These quantities are conserved only in the absence of a background wind; in the presence of a background wind, pseudo-energy and pseudo-momentum are conserved (Nappo, 2013). (Pseudo-)energy or momentum conservation only holds if no energy or momentum dissipation takes place.

**4.1 Ground-based measurements**

Here we select for review a number of OH* airglow atmospheric wave studies utilizing observations from a number of stations or which show long-term results from a comprehensive range of ground based OH* airglow research.





Starting with planetary waves, Reisin et al. (2014) used rotational temperatures measured at 19 sites from 78° N to 76° S to derive monthly and total mean climatologies of planetary wave activity. All of these sites belong to the Network for the Detection of Mesospheric Change (NDMC). Most sites between 69° S and 69° N show similar planetary wave activity; it is only at some high-latitude sites that relatively high activity is observed. At all high-latitude sites, the seasonal variability is relatively high compared to 70% of the mid-latitude stations and the two tropical stations, which show practically no seasonal

variation of PW activity.

Based on 12 years of OH* airglow spectrometer measurements at Wuppertal (51°N, 7°E), Germany, and complementary satellite measurements, Offermann et al. (2009) investigated the influence of planetary and gravity waves as well as tides as sources of fluctuations on vertical temperature profiles in the UMLT. He showed that gravity waves are responsible for the majority of those fluctuations. Travelling planetary waves are more important in this context than quasi-stationary ones. The

strength of tidal fluctuations lies in the range of travelling planetary wave fluctuations or lower depending on latitude (the higher the latitude the lower the tidal strength).

Gravity waves and tides are not easy to separate in OH* airglow measurements since gravity waves can have periods in the range of the semi-diurnal tide, for example, and large-scale spatial information is often not available, which would help to distinguish the different wave types. Therefore, OH* airglow studies of atmospheric tides (relying on time series of some

years or a network of instruments) are not common. One example based on measurements at the Sierra Nevada Observatory (37.06° N, 3.38° W) in the time period of 1998–2003 is given by López-González et al. (2005) who analysed the data with respect to diurnal variations. Both, emission rates and rotational temperatures, show diurnal tidal signals with seasonal variation (greater from late autumn to spring than during summer).

Recently, López-González et al. (2020) analysed OH* airglow measurements at the same observatory from 1998 to 2015 with

respect to gravity waves. Gravity wave activity with periods shorter than 3 h shows a maximum in summer and winter. Gravity waves with periods from 3 to 6 h have maximum activity around October–December and a minimum during summer. These cycles can also be observed in the GWPED, which was derived by Wüst et al. (2016) with five technically identical OH* airglow spectrometers from 2011 to 2014 at four NDMC stations (three stations in central and one in Northern Europe). Due to the instrument sensitivity, those authors split the results into periods longer and shorter than 60 minutes. Based on multi-

year measurements at eight different mostly European stations, Sedlak et al. (2020) showed that there exists a gradual transition in gravity wave activity from a semi-annual cycle with a slight primary maximum in summer for very short periods (60 min and less) over a semi-annual cycle with two maxima of equal strength in summer and winter to an annual cycle with a maximum in winter for periods of ca. 400 min.

Vargas et al. (2015) examined gravity wave propagation directions from OH* airglow imager measurements at more than 20

different stations including those with only a few clear measurement nights up to those that had as many as 513 clear measurement nights in a time period of up to seven years. While the stations are distributed all over the world, the longitudinal sector of ca. 25°W to 75°E is covered by only one station (Esrange, Sweden), which due to its high latitude position (68°N) provided data only during the Northern winter. The preferential propagation direction of high-frequency gravity waves is



toward the summer pole and into the residual meridional circulation. While the zonal propagation direction is consistent with
the assumption of stratospheric wind filtering, there is some inconsistency about the meridional propagation direction. This
picture does not change in more recent literature which focuses on two stations in central Europe (Oberpaffenhofen, Germany,
48.09°N, 11.28°E, and Sonnblick, Austria, 47.05°N, 12.96°E, Hannawald et al. (2019)). Superimposed on the preferred
propagation directions from these stations are local variations.

Rourke et al. (2017) analysed data from a scanning radiometer between 1999 and 2013 and looked for the height of origin of
gravity waves. The instrument deployed at Davis Station, Antarctica (68.6°S, 78.0°E) is sensitive to radiation in the range of
1.10–1.65 μm, and covers a small region (24 km × 24 km) of the zenith night sky once per minute. The authors found that on
average only 15% of the detected gravity waves can be reverse ray-traced to the troposphere. Most of the end points were
found within a radius of 300 km of the station. Ca. 45% were not successfully traced back substantially below the airglow
layer.


In a similar fashion to tides, infrasound and turbulence are not often studied using OH* airglow measurements. In the case of
infrasound, Le Dû et al. (2020) identified infrasound signals in the frequency band of those produced by ocean swell in OH
imager data at the observatory Haute-Provence, France. Inchin et al. (2020) modelled infrasonic waves generated by an
analogue of the 2011 M9.1 Tohoku-Oki earthquake. Those authors found out that these waves may be strong enough to drive
fluctuations in UMLT airglow, which could be detectable with ground- and/or satellite-based imagers. Pilger et al. (2013b)
presented examples for airglow measurements of orographic, volcanic and meteorological infrasound signatures based on
spectrometer data. The only publications in this context, which is based on a larger amount of data, is the one by Pilger et al.
(2013a) who used 40 months of spectrometer data and found evidence for acoustic resonance modes.

The investigation of turbulence in OH* airglow measurements is in most cases based on the observation of so-called ripples
(see e.g. Hannawald et al. (2016); Sedlak et al. (2016)). Ripples are wave-like structures with horizontal wavelengths of 5–15
km (Li et al. (2005); Taylor et al. (1995)) or of 20 km at most (Takahashi et al., 1985). They have a short lifetime of 45 min
or less with periods close to 5 min (Hecht, 2004) and are often interpreted as manifestations of Kelvin-Helmholtz instabilities
(dynamical instabilities) or convective instabilities. However, Li et al. (2017) showed based on OH* all-sky imager
measurement at Yucca Ridge Field Station, Colorado (40.7°N, 104.9°W), from September 2003 to December 2005 and
additional measurements that more than half of the observed ripples might not be instability features but wave structures that
are hard to distinguish from real instability features. In this case, the ripples could be related to secondarily generated small-
scale gravity waves (Vadas et al. (2003); Zhou et al. (2002)). Another possibility to analyse turbulence is the observation of
vortices as Sedlak et al. (2016) and Hecht et al. (2021) showed based on case studies of an imager system with a very high
spatial resolution of 17 m per pixel and 25 m per pixel, respectively. In their more recent work, Sedlak et al. (2021) extended
their study to 1.5 years of data which includes some 20 to 30 case studies. With an instrument originally set up for astronomical
observations (NOTCam spectrograph mounted on the Nordic Optical Telescope (NOT) in La Palma) Franzen et al. (2018)





found quasi-periodic structures with minimal horizontal wavelengths of 4.5 m following a Kolmogorov-type energy cascade during breaking.

### 4.2 Airborne measurements

The number of airborne OH* airglow measurements is in general relatively small. To our knowledge, the first airborne measurement was performed by Noxon (1964) during a single night from a jet aircraft flying at 75°W longitude from 55°N to 85°N. The author analysed the dependence of the rotational temperature and the intensity on latitude. Concerning GW, there are only five larger campaigns reported during which such measurements were performed to our knowledge; three between 1993 and 1999 and two between 2014 and 2016.

Anderson et al. (2008) used an aircraft-mounted spectroscopic imager to tomographically reconstruct atmospheric gravity waves, i.e., to estimate the horizontal–vertical airglow structure. The authors illustrate the applicability of their algorithm based on data taken during the ALOHA campaign, which took place in October 1993. During the same campaign, Swenson and Espy (1995) investigated imager measurements of gravity waves in the vicinity of a tropospheric low pressure system. Due to cloud formation this was not possible in most cases or possible only at large zenith angles (which directly affect the FoV and therefore the spatial resolution) for ground-based instruments.

Siefring et al. (2010) amongst others used an airborne camera (900 to 1700 nm) to observe sprites but also airglow during the Energetics of Upper Atmospheric Excitation by Lightning campaign in 1998 (EXL98). Gravity wave activity and therefore density variations were investigated by the authors. Ionization rates are highly dependent on the neutral density and so gravity waves may have an effect on the location and filamentation of sprites and cause brightness variations in elves. In one case, such a correlation was observed.

During the Leonid MAC Campaign, airglow spectra and imaging data of airborne instruments were used to study upper atmospheric conditions. The temporal focus was on a period centred on the Leonid meteor storm of 17/18 November, 1999 as well as on the meteor storm itself. Kristl et al. (2000) found a small but significant enhancement in the OH airglow emission during the peak of the storm. However, the authors admit that they are not certain of the cause.

During the DEEPWAVE campaign (Deep Propagating Gravity Wave Experiment) project over New Zealand and surrounding regions, a strong gravity wave event was observed by Pautet et al. (2019) during June and July 2014 using three airborne OH* airglow imagers. It remained at the same location for at least 4 h, had a dominant horizontal wavelength of ca. 40 km, and its momentum flux per unit mass was together with additional measurements estimated to be at least 320 m²/s². The authors associated these waves with orographic forcing generated by moderate surface flow over a small island terrain.



In January and February 2016, an airborne OH* airglow camera measured during GW-LCYCLE campaign (Gravity Wave Life CYCLE) in Northern Europe. Two of six flights were analysed by Wüst et al. (2019) with respect to gravity waves with horizontal waves between 3 and 26 km. The results were separated for wavelengths shorter and longer than 15 km (bands and 530 ripples). The most mountainous regions were characterized by the highest occurrence rate of wave-like structures, in one flight due to ripples, in one flight due to bands. At least for one flight, the propagation of mountain waves was not possible or at least strongly reduced.

**4.3 Satellite-based measurements**

Ground-based and airborne instruments are usually not able to provide information about the height of the OH* airglow layer. The first investigations of the OH*-emission height (and in some instances the FWHM of the OH* layer) were performed on a case study basis some 35–45 years ago mostly relying on rocket-borne or lidar measurements (Good (1976); Von Zahn et al. (1987); Baker and Stair (1988) and citations therein).

With the launch of WINDII (Wind Imaging Interferometer) on board UARS (Upper Atmosphere Research Satellite) in 540 September 1991, decadal satellite-based and therefore nearly global limb investigations of the OH* airglow emission were possible. WINDII was followed by OSIRIS (Optical Spectrograph and InfraRed Imager System) on board Odin, SCIAMACHY (Scanning Imaging Absorption Spectrometer for Atmospheric Chartography) on board ENVISAT (ENVIronmental SATellite, January 2003–December 2011), SABER on board TIMED (ongoing since December 2001), ISUAL (Imager of Sprites and Upper Atmospheric Lightnings) on board FORMOSAT 2, and MLS (Microwave Limb 545 Sounder) on board EOS-Aura (Earth Observing System). Details about the measurement parameters and the hemispheric coverage are shown in Table 1.

**Table 1 Summarized are the different satellite-based instruments which deliver information about OH and some additional information about satellite launch date, data availability, observed OH transition and if available additional temperature and/or 550 wind measurements from the instrument, which might be useful in the context of atmospheric dynamics. Data are available some time after the launch date depending on the duration of test phases of the satellite and the instruments.**

| Instrument | Satellite | Launch date* | Data until | Latitudinal range | OH transition | Additional notes | References |
|---|---|---|---|---|---|---|---|
| WINDII Wind Imaging Interferometer | UARS Upper Atmosphere Research Satellite | 12th Sept. 1991 | 2003 | 42° in one hemisphere to 72° in the other one, alternating every 36 days | OH(8-3) | Wind, temperature available | Shepherd et al. (1993), Von Savigny et al. (2004) |
| OSIRIS Optical Spectrograph | Odin | 20th Feb. 2001 | Ongoing | Ca. one hemisphere up to 82°S/N at | OH(8-3), OH(9-4), OH(5-1), | | Llewellyn et al. (2003); |



| | | | | maximum, alternating every half-year | OH(3-1) | | Sheese et al. (2014) https://research-groups.usask.ca/osiris/, 28th June 2021 |
|---|---|---|---|---|---|---|---|
| SABER Sounding of the Atmosphere using Broadband Emission Radiometry | TIMED Thermo-sphere Ionosphere Mesosphere Energetics Dynamics | 7th Dec. 2001 | Ongoing | 52° in one hemisphere to 83° in the other one, alternating every 60 days | OH VER @1.6 µm and 2.1 µm, OH(v=3,4,5) & OH(v=7,8,9) | Temperature available (derived from CO2) | Russell Iii et al. (1999) |
| SCIAMA-CHY Scanning Imaging Absorption spectroMeter for Atmo-spheric Char-tographY | ENVISAT Environ-mental Satellite | 1st March 2002 | 2012 | 35°S–65°N, seasonal variation | OH(3-1), OH(8-3), OH(6-2) | Temperature available | Von Savigny (2015); Von Savigny et al. (2004); Von Savigny et al. (2012); Von Savigny (2015) |
| ISUAL Imager of Sprites and Upper Atmospheric Lightnings | Formosat 2 | 20th May 2004 | 2016 | Only for two orbits each day, addressed lat. range varies, in Northern summer ca. 30°N | OH(9-3) | | Rajesh et al. (2009); Nee et al. (2010); https://directory.eoportal.org/web/eoportal/satellite-missions/f/formosat-2, 28th June 2021 |
| MLS Microwave Limb Sounder | EOS-Aura Earth Observing System, Aura = latin for breeze | 15th July 2004 | Ongoing for 30-day measurement each year since 2011  Stop of continuous measurement 2009 | 82°S–82°N | OH @ ca. 2.5 THz (120 µm) | Temperature available (derived near O2 band) | Waters et al. (2006); https://mls.jpl.nasa.gov/eos-aura-mls/data-products/oh, 28th June 2022 |

Satellite measurements can serve as complementary measurements for ground-based and airborne ones. A very nice example of the benefits that can be derived by combining ground- and satellite-based measurements is illustrated in Shepherd et al.





(2020). The scale of the event chosen for study makes good use of the longitudinal separation of the ground stations. The two ground-based datasets were augmented with temperature profiles from the MLS instrument on the Aura satellite, which enabled the investigators to obtain an overall picture of the state of the atmosphere in the polar cap before, during and after the occurrence of a sudden stratospheric warming (SSW), which had some unusual features. The report illustrates the benefits of using multiple datasets in elucidating different aspects of this complex natural phenomenon. While the satellite data provide

a global overview of the event, the ground-based observations of both temperatures and winds allowed the event to be probed with high time-resolution.

Long-term satellite measurements provide information about the OH* airglow layer characteristics and their development (e.g. Von Savigny (2015) and Wüst et al. (2017)). As mentioned in section 2, height, thickness and brightness of the OH* airglow layer are linked. The relationship between the column-integrated volume emission rate and the peak emission height as it is

derived from satellite-based measurements can be used, for example, to infer the OH*-layer height from ground-based measurements alone (e.g., Liu and Shepherd (2006), Mulligan et al. (2009)).

Satellite-based measurements are also valuable for temperature comparisons with ground based sites (e.g., López-González et al. (2007)) and temperature trend analysis over global scales. French et al. (2020), e.g., combined ground- and satellite-based measurements to assess temperature trends. They derived a long-term cooling trend of $-1.2 \pm 0.51$ K per decade based on 24

year of measurements at Davis Station, Antarctica. In their table 1 those authors also provide trend estimates based on further OH* airglow measurement stations from the literature, which are all negative. Additional analyses of 14 years of nearly global AURA/MLS data show that the temperature trends at the hydroxyl layer equivalent pressure level are significantly non-uniform over the globe and that positive trends can be observed in some regions.

Further satellite-based OH* airglow instruments, which should be mentioned in this context are nadir looking ones such as

VIRRS DNB (Day/Night Band nightglow imagery from the Visible/Infrared Imaging Radiometer Suite) on board Suomi NPP (Suomi National Polar orbiting Partnership, ongoing since October 2011) and JPSS-1 (Joint Polar Satellite System-1). They do not provide additional information about the OH* layer height or thickness such as limb viewing systems, however they allow for the analysis of horizontal GW parameters with a spatial resolution of ca. 0.75 km on moonless nights independent of the weather conditions (see Miller et al. (2015) and Yue et al. (2019) and references mentioned therein). This is of great

advantage for the observation of gravity waves generated by convective systems (e.g., Miller et al. (2012), Xu et al. (2019), Yue et al. (2014)), which is limited from the ground due to cloud coverage. The use of nadir measurements is not restricted to this purpose, but includes other possible GW sources such as orography or seismic and volcanic events that are investigated (Miller et al., 2015). This is done in case studies. Long-term studies relying on VIRRS DNB are yet to be published to our knowledge.




## 5 Summary and outlook

Ground-based OH* airglow measurements have been carried out for almost 100 years. At some sites they are available for decades. Advanced detector technology has greatly simplified the automatic operation of OH* airglow observing instruments and significantly improved the temporal and, in the case of imagers, the spatial resolution of these measurements. Studies

based on long-term measurements (i.e., ten years or more) or including a network of instruments have been reviewed especially in the context of deriving gravity wave properties.

The parameters investigated include periods or horizontal wavelengths, which always depend on the technical specifications and the measurement geometry of the instruments. Also wave propagation direction and activity (which is defined differently in many studies) or density of potential energy are derived. Today's challenges lie to a large extent in the field of analyses.

Improved temporal and spatial resolution reduce averaging effects in space and time, therefore, more non-stationary processes become evident in the data. Additionally, the amount of data increases.

Information about the wind at the OH* airglow layer height allow the computation of additional wave parameters. Wind data based on the OH* airglow can be measured with Fabry-Perot or Michelson interferometers. However, those measurements are not quite frequent.

Ray tracing studies for source attribution of waves measured through OH* airglow observations and whether these waves are primary ones (so generated directly by convection or flow over mountains, for example) or result from possible wave-wave or wave-mean flow interactions are rare. For this, additional information e.g., on the state of the atmosphere (especially temperature, wind) at other altitudes are needed. Wave propagation models can provide further information, if the corresponding processes are included. For the analysis of non-linear wave-wave interactions, comprehensive information about

the parameters of the different waves are needed (see e.g. Wüst and Bittner (2006)).

The horizontal resolution of ca. 10–20 m, which is achieved today in the best case by OH* airglow imagers, makes it possible to observe turbulence eddies and to estimate the energy emitted by gravity waves and thus the heating rates of the atmosphere by gravity waves. However, for the analysis of larger data sets, the automatic detection of these vortices is necessary. This requires the use of other analysis methods than those applied so far. Here, first steps are currently being taken in the field of

artificial intelligence and machine learning. Lai et al. (2019) showed, for example, how to identify cloudless nights in all-sky OH*-airglow imager time series and how to locate the wave patterns in each picture.

The observation of infrasound with the help of OH* airglow measurements is currently still quite difficult. Especially in the context of observing and learning more about natural hazards (Bittner et al., 2010), the detection of infrasound is of interest as Inchin et al. (2020) modelled. Here, OH* airglow intensity measurements are better suited than temperature observations.

Clouds generally hinder the ground-based detection of OH* airglow emissions, sunlight significantly reduces the detected airglow intensity. For other species such as oxygen airglow (green line: 557.7 nm centred at night around 97 km+/- 3 km height (Wolff, 1966), red line: 630 nm centred around 300 km (Danilov, 1962), which is relatively broad compared to the other airglow lines mentioned here), it could be shown that ground-based daytime measurements are possible based on different



instruments (see Marshall et al. (2011), Pallamraju et al. (2002) and citations therein). For OH* airglow, at least to our
knowledge only one technique is published using a multiwavelength photometer (Sridharan et al., 1998). OH* dayglow shows
a double-peak structure: the upper layer is located at a similar altitude as in the case of OH nightglow, the lower layer is
between 70 km and 85 km height increasing with time of the day (Gao et al., 2015). Additionally, the proportion of scattered
light is relatively high during day, so that very narrow-band filters are necessary, which reduces the light yield and decreases
the temporal resolution. Daytime measurements of OH* airglow therefore do not allow a comparable continuation of nocturnal
measurements.

**Author contribution**

SW defined the subject of the manuscript, did the literature search and wrote the first version of the manuscript. It was discussed
with all co-authors resulting in the present form of the manuscript.

**Acknowledgement**

The work of Sabine Wüst was funded by the Bavarian State Ministry for the Environment and Consumer Protection (VoCaS-
ALP, TKP01KPB-70581, and WAVE, TKO01KPB-73893).
We thank Jürgen Scheer for the valuable discussion.

**Competing interest**

The authors declare that they have no conflict of interest.



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
