# Peer review of "Hydroxyl airglow observations for investigating atmospheric dynamics: results and challenges"

_Atmospheric Chemistry and Physics, 2022_

## Referee Comment (RC2)

Review of the paper: "Hydroxyl airglow observations for investigating atmospheric dynamics: results and challenges" by Wüst et al. .

General comment.

This is excellent review paper. It can be excepted for publication in Atmospheric Chemistry and Physics after very minor correction which may take into an account comments of other referees. I have just several minor comments.

Specific comments.

1. Introduction, $1^{st}$ paragraph. Additionally were conducted number of rocket-borne measurements, for example MULTIFOT 92 (Takahashi et al., 1996).

2. Page 2. "Both parameters can vary (in the case of the centroid height by some kilometers) over several days or even during a single night due to strong dynamics (e.g., changes in the residual circulation during a stratospheric warming or strong tidal motions)". – Some modelling results show monthly averaged variation from ~78 km to ~91 km and in frame of single month it can be even stronger, specifically at high latitudes (Grygalashvyly et al., 2014). This is much complex question than briefly noted here, because altitude variation of OH* layer depends on latitude, season, vibrational number. From the other hand this is just introduction and should not cover all problems. Hence, authors may slightly extend this discussion or not, on their choice.

3. Page 2. "adjacent vibrational levels are separated by some 100 m (Baker and Stair, 1988; Adler-Golden, 1997" it seems to me that a little bit stronger.
Backer and Stair (1988) found ~ 500 m, Adler-Golden (1997) found ~700 m, Grygalashvyly et al. (2014) found 250-1000 m depending on season and latitude.

4. Page 3. The abbreviation MORTI should be disclosed.

5. Page 5, Section 2.1. Reaction 2 was proposed as hypothesis, which has not been confirmed and currently it is not considered.

6. Page 5, Section 2.1. Reaction 3. The reaction $O+HO_2 \rightarrow OH^*+O_2$ was introduced as a source of vibrationally excited hydroxyl in the 1970s by, probably (?), Nagy et al. (1976) as a hypothesis put forward for energy reasons (the energy of this exothermic reaction is sufficient to produce OH* up to and including the $6^{th}$ vibrational level) and was applied by several authors in the 1980s to explain discrepancies between observed emissions and calculation results (Takahashi and Batista, 1981; Turnbull and Lowe, 1983). At that time there were no sufficiently good measurements and calculations of molecular and atomic oxygen quenching coefficients, spontaneous emission coefficients and yield coefficients of the reaction of ozone with atomic hydrogen. But already Llewellyn et al. (1978) noted that with the new quenching coefficients they calculated, a new OH* source (R3) would no longer be necessary. Further, Kaye (1988) showed from laboratory measurements that population above the $3^{rd}$ vibrational level is not possible. Moreover, population coefficients for the first three levels have been proposed (Makhlouf et al., 1995) using general considerations without solid confirmations. To date, no more precise information on the exit coefficients has been obtained. Furthermore, with new calculated and laboratory-derived quenching coefficients, spontaneous emission coefficients, and yield coefficients for the ozone and hydrogen atom reaction, the application of hydroperoxide and oxygen atom reaction to obtain agreement on OH* emission measurements is not required (Xu et al., 2012; McDade and Llewellyn, 1987). Although some authors still apply this reaction, it can be omitted from consideration until the time when it will be supported based on laboratory measurements.

If discussion in Section 2.1. cover day and night conditions, may be for authors will be interesting that recently was shown that water vapour dissociation may essential contribute to daytime OH* population.

Ch a n g , Y . , Li, Q. M., An, F., Luo, Z. J., Zhao, Y., Yu, Y., He, Z., Chen, Z., Che, L., Ding, H., Zhang, W., Wu, G., Hu, X., Xie, D., Plane, J. M. C., Feng, W., Western, C. M., Ashfold, M. N. R., Yuan, K., & Yang, X. (2020). Water photolysis and its contributions to the hydroxyl dayglow emissions in the atmospheres of Earth and Mars. Journal of Physical Chemistry Letters, 11, 9086-9092. https://doi.org/10.1021/acs.jpclett.0c02803

7. Page 5, Section 2.1. "Lower levels are populated in a radiative cascade by spontaneous emission" – I would not use the word "cascade" because it is related to "cascade" scheme at which the excited molecule relaxes to the one vibrationally excited level below (e.g. McDade

and Llewellyn, 1987), but in nature all quenching and spontaneous emission processes are neither "cascade" nor "sudden death", but of multi-quantum relaxation nature.

8. Figure 1b, left panel. Why the nighttime atomic oxygen concentration, taking into an account discussion above, is higher than that for daytime at ~80-85 km?

9. Figure 1b, right panel. Currently well known that OH*-layers with higher vibrational numbers have peaks higher than those for smaller vibrational numbers (e.g., McDade, 1991; Adler-Golden, 1997; and references therein). On this figure the sequence of vibrational numbers from higher altitude downwards is as follow: 5-4-7-6-3-2.
Why it is?

The number density of OHv at peak grows in the direction of the smaller $v$ (e.g., Sivjee and Hamwey, 1987; McDade, 1991; Adler-Golden, 1997; Xu et al., 2012; Caridade et al., 2013) On this figure the sequence of vibration numbers from lower concentrations toward higher is as follow: 6-5-4-7-3-2.
Why it is?

On my opinion this is not the best illustration.

All of the above are non-binding corrections to the article and are left to the authors' discretion. All references mentioned in the review are not required to be cited in the article. Generally, after specific and technical corrections, I recommend this paper for publication in Atmospheric Chemistry and Physics.

---

## Author Comment (AC1)

Thank you, Christian, for your valuable comments. They improve the manuscript. Below you find my answers. In the majority of cases I changed the manuscript accordingly.

General comments:

This is a generally well written paper on the exploitation of measurements of the OH Meinel airglow emissions with a focus on studying dynamical processes in the mesopause region. The paper does not really contain any new science results, but rather has the character of a review paper. In my opinion it is a useful contribution to the field, because it provides background information that is usually not included in that detail in more specific papers.

I have one general comment: I'm a bit surprised that the dependence of derived rotational temperatures on the vibrational excitation (works by Noll et al.) is not really mentioned in this paper. The paper discusses several potential issues regarding the measurement of temperatures from the OH emissions and this important aspect is not addressed. I suggest adding a brief discussion on the current level of understanding of the dependence of retrieved temperatures on the vibrational level.

I included some information about this topic in the revised manuscript (section 3, derivation of temperature)

Below I offer some comments and suggestions for improvements for the authors to consider.

Specific comments:

One point from my side concerning the Skycorr model mentioned in the introduction and described in a footnote: I got an email from Stefan Noll, the lead author of this model; he told me that the webpage to which we are referring in the manuscript describes the Skycalc model. Even though both models are related (Skycorr subtracts the airglow emission, Skycalc calculates it), the correct nomenclature is Skycalc. Therefore, I changed it.

This is only minor comment, but the abstract appears a bit short. All relevant topics seem to be covered, but perhaps you want to work out the abstract a bit more?

I extended it with respect to section 2 and 3 and provided some more infos about section 4.

Line 23: "At present, nocturnal hydroxyl (OH*) airglow measurements are performed from the ground, aircraft, balloon and space." There were also some ship-based measurements (on Polarstern) by the group of the author, right? Perhaps this should be mentioned, too. In addition, I'm not sure, if the tense of the sentence is correct. These measurements are not all

performed right now or these days, right? I guess, you want to express that these measurement platforms have been used in the past?

I included the ship measurement and changed the tense to past present perfect, I suppose that's the correct one.

Line 24: "1950's" -> "1950s"

Corrected

Line 25: "More recently, satellite-based and airborne measurements have been carried out that are not severely affected by clouds in most cases." Well, satellite measurements are not affected by clouds at all, right?

Yes, this part of the sentence should refer only to the airborne measurements. I corrected it accordingly ("More recently, satellite-based and airborne measurements have been carried out. The first ones are never and the latter ones are not severely affected by clouds. Satellites provide global or nearly global data, airplanes the greatest spatial flexibility.").

Line 35: "Some of the transitions with low rotational quantum numbers are approximately in local thermodynamic equilibrium". Wouldn't we say that the rotational population is in LTE, rather than stating that the transitions are in LTE?

Yes, that's true, I changed it.

Line 50 (and a couple of other lines): "Von Savigny" -> "von Savigny"

Here I was fighting with the citation program, finally I changed it manually (that hold's also for Russell Iii and Lednyts'Kyy). As we are speaking about the citation program, I realized that the literature mentioned in the footnotes is not provided in the literature list. I changed this, too.

Lines 81 – 85: I suggest also citing the corresponding instrument papers (i.e. Russell et al., Llewellyn et al. (2004) (Llewellyn et al., Can. J. Phys. (2004) is probably better than Llewellyn et al. (2003), see below), Waters et al.)

Added

Line 89: „the main points" -> „the main processes" ?

Corrected

Line 113: "Lower levels are populated in a radiative cascade by spontaneous emission" or by collisional relaxation?

I inserted "or by collisional relaxation as addressed later by equation (6)" after the chemical equation.

Line 138: "The produced OH* has a very short lifetime," Perhaps you can provide a number here? Some people may think that you mean typical lifetimes on the order of $10^{-10}$ s or so.

Same line: "it takes less than 1 s before it relaxes through collision" I suggest explicitly mentioning a typical value of the collisional frequency at around 87 km altitude.

I inserted "The Einstein coefficient for the transition from the 9th to the 7th or the 8th to the 6th vibrational levels, for example, are larger than $10^2$ s$^{-1}$ (Marsh et al., 2006), so the lifetime of the corresponding state is smaller than $10^{-2}$ s. A summary of the majority of published Einstein coefficients for the different OH vibrational transitions is available in Table 2.3 – 2.13 in Khomich et al. (2008)" as footnote and provided the collisional frequency in brackets in the main text.

Line 148: One might also cite von Savigny & Lednytskyy (GRL, 40(21), 5821 – 5825, 2013) providing experimental evidence on the importance of O.

Done – even though Word doesn't show it as changed. Changes in the text which are made by the citation program Endnote, which I used, are not marked by Word.

Figure 1: I suggest mentioning in the figure caption that a) shows nighttime conditions

The figure is a reproduction of Swenson and Gardner (1998). They don't provide any info about day or night. Compared to part b) it is not clear to me whether part a) represents clearly night or day conditions since part a) ist only restricted to 80 – 100 km height. Therefore, I would prefer leaving out the info about day or night here.

Line 193: "A semi-annual cycle is also often found" At low latitudes the semi-annual variation is dominant and there is no doubt it is present! See, e.g. von Savigny & Lednytskyy (2013) and also many other studies on this topics. But I completely agree that the interpretation problem is present at mid and high latitudes.

Thanks for the clarification, I changed the sentence accordingly ("It is either attributed to physical mechanisms especially at low latitudes (von Savigny and Lednyts' kyy, 2013) or treated as an analysis artefact, which is necessary to compensate for the fact that the annual cycle does not correspond exactly to a sine wave.")

Figure 2: I hadn't heard about these factors before and didn't understand from the explanations in this paper what they exactly are. Please explain in more detail, if possible. Why are they negative?

Swenson and Gardner (1998) showed that the perturbed rotational temperature $\Delta T_R$ and the perturbed volume emission rate $\Delta V$ relate to the unperturbed volume emission rate $V_u$ and to the perturbed and unperturbed densities, $\Delta\rho$ and $\rho_u$, as follows:

$$\Delta T_R(z) \approx g_T(z)\, V_u(z)\Delta\rho/\rho_u$$

$$\Delta V(z) \approx g_{OH}(z)\, V_u(z)\Delta\rho/\rho_u$$

(formulas (31) and (40) of Swenson and Gardner (1998)). The g-term in combination with the unperturbed OH* profile $V_u(z)$ is denoted by those authors as a weighting factor, which describes the relative contribution of the wave perturbations $\Delta\rho/\rho_u$ to the perturbed temperature (volume emission rate) versus altitude.

A wave propagating through the OH*-layer influences amongst others, O, $O_2$, and temperature which change the volume emission rate. $g_{OH}$ includes those contributions.

Fluctuations of the temperature profile, the centroid height and the mean square width of the OH-layer are included in $g_T$.

Both g-factors relate the corresponding perturbations to the density perturbations. Being negative means that the perturbations of T and V are not in phase with the density perturbation in my opinion.

Line 240: „... the relation .. need to be ..“ -> „... the relation .. needs to be ..“

 Corrected

Line 242: „In general, the lowest rotational transitions of the lower OH* vibrational transitions are sufficiently close to LTE“ Same question as above: are the transitions in LTE or rather the rotational population?

Changed

Line 249: „two primed variables“ -> „double primed variables“ ? But the native speakers among the authors will know this better than I do.

Even though, my co-authors were raised in different English speaking countries, the answer was unambiguous: it's double primed and I changed it accordingly.

Line 252: „If LTE holds for the rotational transitions“ -> "If LTE holds for the rotational populations“ ?

Changed

Equation (8): again, only a minor point, but if the rotational population is in LTE, then T_rot = T_kin. Perhaps this can be mentioned.

Yes, I changed it. Additionally, I corrected formula (10) – there, only T not $T_{rot}$ was mentioned – and I added the number of this equation two lines below where there was only a pair of brackets but no number in it.

Line 275: „In many cases, three or more rotational lines are used, which makes the fit more robust, and provides an immediate measure of the uncertainty of the temperature retrieved (e.g., French et al. (2000), Sigernes et al. (2003)).“ The problem with the non-LTE population of the higher rotational states could be mentioned here, as it affects the derived temperatures.

I included "Based on astronomical sky spectra, Cosby and Slanger (2007) and Noll et al. (2017), for example, showed that retrieved temperatures T_rot depend on the vibrational level v^'. There is the overall trend that higher vibrational levels lead to higher temperatures (non-LTE effect) even though there is some variation with the vibrational level. This observation is justified by the thermal equilibration efficiency by collisions which is altitude-dependent (and therefore v^'-dependent)."

Line 283: „Therefore, the free spectral range is larger than for the Q- and R-branches" What does „free spectral range" mean here exactly? I only know this expression in the context of transmission functions of, e.g. etalons.

I mean the distance between two lines in the spectrum and added this info in parentheses in the manuscript.

Line 298: „Comparison of result" -> „Comparison of results"

Corrected

Line 326 and line 329: „based on a single detector cells" -> „based on a single detector cell"

Corrected

Line 327: „The size of the FoV depends on the optics of the spectrometer but also on the zenith angle at which the instrument is operated." Well, the FoV (given as a solid angle) is independent of the zenith angle, right? I guess you mean the size of the sensed air volume / area at OH layer altitude?

Yes, concretized.

Line 329: „in the context of satellite validation the reduced sensitivity" What does „reduced sensitivity" refer to here? This is not entirely clear to me.

I placed this sentence behind the next one and changed the wording slightly (in the context of satellite validation the reduced sensitivity to spatial features due to the non-infinitesimal small sensed air volume is known as observational filter).

Line 337: „The only publication reporting a temperature imaging system, so not a scanning one, is by Pautet et al. (2014)" The SATI measurements (also using OH emissions) have some imaging (although limited) imaging capability, too.

Yes, that's true, thanks, I inserted the info ("When using OH* airglow cameras with suitable filter wheels which are sensitive to different rotational transitions of one vibrational band, temperature information can be retrieved (without scanning the night sky) as done by Pautet et al. (2014); also SATI provides a temperature image but with a worse spatial resolution than Pautet et al. (2014) covering 12 different azimuth angles of the observed annulus in the airglow layer (López-González et al., 2007)").

Line 415: space betwen \lambda_z and „and" missing.

Corrected

Line 438: „He showed" -> „They showed"

Corrected

Line 442: „Gravity waves and tides are not easy to separate in OH* airglow measurements" That's an interesting point. I would have thought that tidal variations will be similar on subsequent days. Then a tidal variation could be extracted with a composite (or superposed

epoch) analysis applied to many nights in a month or a season. The paper by Shepherd & Fricke Begemann (2004) comes to my mind, where tidal temperature variations based on K lidar measurements were determined: Shepherd, M. and Fricke-Begemann, C.: Study of the tidal variations in mesospheric temperature at low and mid latitudes from WINDII and potassium lidar observations, Ann. Geophys., 22, 1513–1528, https://doi.org/10.5194/angeo-22-1513-2004, 2004.

When I wrote this sentence, I had Silber et al. (2017) in my mind (Silber I., Price, C., Schmidt, C., Wüst, S., Bittner, M., Pecora E., 2017. First ground-based observations of mesopause temperatures above the Eastern-Mediterranean Part I: Multi-day oscillations and tides. Journal of Atmospheric and Solar-Terrestrial Physics, 155, 95-103, DOI: 10.1016/j.jastp.2016.08.014). Those authors analysed airglow spectrometer (GRIPS) data from Tel Aviv also with respect to tides. They used GRIPS data of four years and applied a spectral analysis consecutively to sub time series of seven days (shifted by one day each time). In their figure 7, they showed the amplitude variation of the different tides with time. Based on this figure, I would say that one can only assume similar amplitudes over very few consecutive days. Since OH measurements depend on clouds, these consecutive nights might not be available.

I changed the part starting with „Gravity waves and tides are not easy to separate in OH* airglow measurements …" to "It can be a challenge to separate gravity waves and tides in OH* airglow measurements since gravity waves can have periods in the range of the semi-diurnal tide, for example, and large-scale spatial information is often not available, which would help to distinguish the different wave types. Another possibility to separate gravity waves and at least migrating tides is to search for signals with phases (see, for example, figure 6 of Silber et al. (2017)) and periods typical for tides over some consecutive nights. However, data availability has to allow for this procedure. For the extraction of tides based on a spectral analysis, for example, it is worth keeping in mind that tidal amplitudes can change within a few days (Silber et al., 2017). Probably due to these reasons, OH* airglow studies of atmospheric tides (relying on time series of some years or a network of instruments) are not common."

Line 464: „and into the residual meridional circulation" I'm not sure I understand what this means?

I reformulated this part: "to the East (West) during summer (winter)"

Line 496: „Franzen et al. (2018)" is not listed in the reference list.

Inserted

Line 539 and following: GOMOS on Envisat was also used to study OH emissions, e.g.: Bellisario et al., O2 and OH night airglow emission derived from GOMOS-Envisat instrument, Journal of Atmospheric and Oceanic Technology, 31(6):1301-1311, 2014.

Thanks for the hint, I inserted the info.

Same text block: perhaps the instrument papers should be mentioned here?

Provided

Table 1, WINDII line: reference to von Savigny is misplaced here.

Corrected

Table 1, SABER line: "Russell Iii" -> "Russell III"

Corrected

Table 1, SCIAMACHY line: SCIAMACHY operations ended on April 8, 2012.

Inserted

Same line: "von Savigny (2015)" is listed twice. Thanks for the honour, but one time is sufficient.

Changed

Line 617: "which is relatively broad compared to the other airglow lines mentioned here" Why is it broader? There are two lines (630 nm and 636 nm), but each line is not really broader than the green line, I think. I checked GLO spectra by Lyle Broadfoot and the individual red lines don't appear to be broader than the green line.

That was a really misleading formulation. I meant not the line itself but the thickness of the layer. I changed "lines" to "layers".

Llewellyn et al. (2003): this paper certainly also deals with the OSIRIS instrument, but the „official" OSIRIS paper ist the following: Llewellyn, E. J., N. D. Lloyd, D. A. Degenstein, R. L. Gattinger, S. V. Petelina, A. E. Bourassa, J. T. Wiensz, E. V. Ivanov, I. C. McDade, B. H. Solheim, J. C. McConnell, C. S. Haley, C. von Savigny, C. E. Sioris, C. A. McLinden, E. Griffioen, J. Kaminski, W. F. J. Evans, E. Puckrin, K. Strong, V. Wehrle, R. H. Hum, D. J. W. Kendall, J. Matsushita, D. P. Murtagh, S. Brohede, J. Stegman, G. Witt, G. Barnes, W. F. Payne, L. Piche, K. Smith, G. Warshaw, D.-L. Deslauniers, P. Marchand, E. H. Richardson, R. A. King, I. Wevers, W. McCreath, E. Kyrölä, L. Oikarinen, G. W. Leppelmeier, H. Auvinen, G. Megie, A. Hauchecorne, F. Lefevre, J. de La Nöe, P. Ricaud, U. Frisk, F. Sjoberg, F. von Scheele, and L. Nordh, The OSIRIS instrument on the Odin satellite, Can. J. Phys., 82, 411 – 422, doi:10.1139/P04-005, 2004

Added

**Answer to the comments of referee 2 (RC 2):**

We would like to thank the anonymous reviewer for his / her valuable comments and additional information. I found them very helpful and changed the manuscript accordingly. However, I didn't provide the whole amount of information but condensed them a little bit.

**General comment.**

This is excellent review paper. It can be excepted for publication in Atmospheric Chemistry and Physics after very minor correction which may take into an account comments of other referees. I have just several minor comments.

**Specific comments.**

One point from my side concerning the Skycorr model mentioned in the introduction and described in a footnote: I got an email from Stefan Noll, the lead author of this model; he told me that the webpage to which we are referring in the manuscript describes the Skycalc model. Even though both models are related (Skycorr subtracts the airglow emission, Skycalc calculates it), the correct nomenclature is Skycalc. Therefore, I changed it.

Another point: I used the citation program endnote. When adding a citation in Word (tracking changes mode) with Endnote it is not marked as inserted. I am sorry for that.

1. Introduction, 1st paragraph. Additionally were conducted number of rocket-borne measurements, for example MULTIFOT 92 (Takahashi et al., 1996).

    Info included

2. Page 2. "Both parameters can vary (in the case of the centroid height by some kilometers) over several days or even during a single night due to strong dynamics (e.g., changes in the residual circulation during a stratospheric warming or strong tidal motions)". – Some modelling results show monthly averaged variation from ~78 km to ~91 km and in frame of single month it can be even stronger, specifically at high latitudes (Grygalashvyly et al., 2014). This is much complex question than briefly noted here, because altitude variation of OH* layer depends on latitude, season, vibrational number. From the other hand this is just introduction and should not cover all problems. Hence, authors may slightly extend this discussion or not, on their choice.

    I mentioned that variations show also latitudinal and seasonal dependence.

3. Page 2. "adjacent vibrational levels are separated by some 100 m (Baker and Stair, 1988; Adler-Golden, 1997" it seems to me that a little bit stronger. Backer and Stair (1988) found ~ 500 m, Adler-Golden (1997) found ~700 m, Grygalashvyly et al. (2014) found 250-1000 m depending on season and latitude.

   I explicitly mention that the upper limit of 1 km & provided the additional citation (however, word is not marking new citations made with Endnote, the citation program I use).

4. Page 3. The abbreviation MORTI should be disclosed.

   Info included.

5. Page 5, Section 2.1. Reaction 2 was proposed as hypothesis, which has not been confirmed and currently it is not considered.

   Text reformulated.

6. Page 5, Section 2.1. Reaction 3. The reaction $O+HO_2->OH^*+O_2$ was introduced as a source of vibrationally excited hydroxyl in the 1970s by, probably (?), Nagy et al. (1976) as a hypothesis put forward for energy reasons (the energy of this exothermic reaction is sufficient to produce OH* up to and including the $6^{th}$ vibrational level) and was applied by several authors in the 1980s to explain discrepancies between observed emissions and calculation results (Takahashi and Batista, 1981; Turnbull and Lowe, 1983). At that time there were no sufficiently good measurements and calculations of molecular and atomic oxygen quenching coefficients, spontaneous emission coefficients and yield coefficients of the reaction of ozone with atomic hydrogen. But already Llewellyn et al. (1978) noted that with the new quenching coefficients they calculated, a new OH* source (R3) would no longer be necessary. Further, Kaye (1988) showed from laboratory measurements that population above the $3^{rd}$ vibrational level is not possible. Moreover, population coefficients for the first three levels have been proposed (Makhlouf et al., 1995) using general considerations without solid confirmations. To date, no more precise information on the exit coefficients has been obtained. Furthermore, with new calculated and laboratory-derived quenching coefficients, spontaneous emission coefficients, and yield coefficients for the ozone and hydrogen atom reaction, the application of hydroperoxide and oxygen atom reaction to obtain agreement on OH* emission measurements is not required (Xu et al., 2012; McDade and Llewellyn, 1987). Although some authors still apply this reaction, it can be omitted from consideration until the time when it will be supported based on laboratory measurements.

   Thanks for the overview. I changed the text accordingly ("Today, at least equation (1) is considered. Additionally, some authors also take or took equation (3) into account (see, for example, Snively et al. (2010) and Grygalashvyly et al. (2014)). Its necessity but also the vibrational levels which can be populated by this reaction are subject of some discussion as outlined, for example, by Xu et al. (2012) in their introduction.")

If discussion in Section 2.1. cover day and night conditions, may be for authors will be interesting that recently was shown that water vapour dissociation may essential contribute to daytime OH* population.

C h a n g , Y . , Li, Q. M., An, F., Luo, Z. J., Zhao, Y., Yu, Y., He, Z., Chen, Z., Che, L., Ding, H., Zhang, W., Wu, G., Hu, X., Xie, D., Plane, J. M. C., Feng, W., Western, C. M., Ashfold, M. N. R., Yuan, K., & Yang, X. (2020). Water photolysis and its contributions to the hydroxyl dayglow emissions in the atmospheres of Earth and Mars. Journal of Physical Chemistry Letters, 11, 9086-9092. https://doi.org/10.1021/acs.jpclett.0c02803

The discussion is focused on night conditions but thanks for the hint.

7. Page 5, Section 2.1. "Lower levels are populated in a radiative cascade by spontaneous emission" – I would not use the word "cascade" because it is related to "cascade" scheme at which the excited molecule relaxes to the one vibrationally excited level below (e.g. McDade and Llewellyn, 1987), but in nature all quenching and spontaneous emission processes are neither "cascade" nor "sudden death", but of multi-quantum relaxation nature.

I deleted "cascade". Due to the given chemical reaction it should become clear now that a one-by-one "classical" cascade is not meant.

8. Figure 1b, left panel. Why the nighttime atomic oxygen concentration, taking into an account discussion above, is higher than that for daytime at ~80-85 km?

[Figure]

FIG. 1. H, $O_3$, O AND M PROFILES.
Solid lines are noon profiles; dashed lines, midnight profiles.

This 'feature' is in the original figure from their paper (see figure above). They are derived from a time-dependent one-dimensional model and it is noted that the rate coefficients have considerable uncertainty. There is a discussion of the deactivation of OH* by O during the day, compared to nighttime starting in the LHS bottom paragraph on pg 1423 of their paper but not specifically why the daytime O concentration may dip lower than nighttime at ~82km in their model.

Smith et al (2010) have examined the diurnal cycle in [O] from SABER observations. Their figure 2 (and table 1) does show a similar pattern of lower [O] during the daytime ~85km which they describe in the text (page 6-7) as small but no further explanation.

[Figure]

**Figure 2.** Profiles of global mean atomic oxygen averaged over all latitudes for 7.6 years. Solid is night-time; dashed is daytime. Units are (left) volume mixing ratio and (right) number density (cm$^{-3}$).

Their figures 3 and 6 also shows a midnight peak in O mixing ratio between 82 and 84km from 7.6 years of averaged SABER profiles.

[Figure]

**Figure 3.** Local time variation of atomic oxygen at specified pressure levels derived from SABER emissions (black) and predicted using the assumption that variations are due to vertical transport (red) averaged over the latitudes 20°S–20°N. Units are volume mixing ratio ×10$^3$.

[Figure]

**Figure 6.** Local time variation of atomic oxygen at all latitudes at four pressure levels averaged over 7.6 years. Units are volume mixing ratio ×10⁵.

So, this 'feature' does not seem to be the result of uncertainty in time-dependent rate coefficients in that early 1D model of Battaner and Lopez-Moreno but there does seem to be observational evidence that there is a midnight peak in [O] particularly near the equator.

In the manuscript, we added the citations Xu et al (2012) and Smith et al (2010). Furthermore, we would like to point out that there is given a hint concerning the large uncertainties of O in the caption of figure 1.

9.  Figure 1b, right panel. Currently well known that OH*-layers with higher vibrational numbers have peaks higher than those for smaller vibrational numbers (e.g., McDade, 1991; Adler-Golden, 1997; and references therein). On this figure the sequence of vibrational numbers from higher altitude downwards is as follow: 5-4-7-6-3-2.

Why it is?

 The number density of OHv at peak grows in the direction of the smaller *v* (e.g., Sivjee and Hamwey, 1987; McDade, 1991; Adler-Golden, 1997; Xu et al., 2012; Caridade et al., 2013) On this figure the sequence of vibration numbers from lower concentrations toward higher is as follow: 6-5-4-7-3-2.

Why it is?

On my opinion this is not the best illustration.

Again this panel is reproduced from Fig 2 in Lopez-Moreno. The original is black and white and a bit difficult to discern the dash – dot patterns for each v'. In this early work

these profiles were derived from observations from a single rocket flight (19-Dec-1981) sampling in three OH channels, from which the v' profiles were obtained. Not surprisingly they are approximate, however their conclusion was that "The individual profiles show that there is a dependence of the characteristics of the layers (altitude of the peak and half width) on the vibrational level. The lower levels extend to lower altitude than the higher ones" which is all that we really want to show here in relation to the concentration of the reacting species in panel b).

Nevertheless, from latter studies, the higher v' profiles in this figure are out of place. We replaced this figure with a reproduction of figure 1 of Xu et al., (2012)

All of the above are non-binding corrections to the article and are left to the authors' discretion. All references mentioned in the review are not required to be cited in the article.

Generally, after specific and technical corrections, I recommend this paper for publication in Atmospheric Chemistry and Physics.

---

## Author Response (AR2)

Dear Bernd Funke,

thanks for the hint – yes, ENVISAT is the satellite. I corrected the table, the corresponding paragraph in the text was already ok.

For the production process, I additionally adapted the labelling of the x- and y-axis of figure 1a to figure 1b. The axes of both figures show the same but synonyms were used for the labelling. Furthermore, I adjusted the fonts and the sizes of the fonts.

I have one question concerning figure 1: Both parts are drawn by one of my co-authors and me according to the mentioned publications. Do I need any rights from the respective journals?

Best regards, Sabine Wüst